# Learning an Inventory Control Policy with General Inventory Arrival Dynamics

## Abstract

We apply deep reinforcement learning (RL) to solve the periodic review inventory control problem with general arrival dynamics. In this work, we incorporate a learned model of transition dynamics (inventory arrivals) into the inventory control problem formulation, increasing the fidelity of the resulting simulator. Leveraging recent results (Madeka et al., 2022), we demonstrate a reduction of the complexity of the inventory control problem we consider to that of supervised learning, proving that under mild assumptions our backtest of inventory control policies is accurate. We also propose several metrics by which to evaluate the inventory arrivals model, and demonstrate the impact of an improved arrivals model on policy performance via a comparison of policies learned on our simulator with one learned on a simulator with less accurate arrivals dynamics. Finally, we use data from a real world A/B test of an RL agent trained using our simulator with learned dynamics to evaluate the performance of the arrivals model, showing that empirically it generalizes well to the *off-policy* state distribution induced by the RL agent in an actual supply chain.

## 1 Introduction

Many applied optimization problems such as inventory control, staff scheduling, route planning, and resource allocation involve some sort of decision that a planner takes in order to maximize some notion of reward (or, minimize cost), which is realized from the environment. Traditional operations research methods have typically been employed as approximate solutions to these problems, by making the problem more tractable with simplifying assumptions on the problem dynamics. Recently, much interest has gathered around using deep reinforcement learning (RL) to dynamically learn the best decision-making policy through direct interactions with the environment (real or simulated). In general, despite promising applications of RL to fields such as game-play (Atari, AlphaGo, Starcraft) and robotics (Mnih et al., 2013; Silver et al., 2016; Vinyals et al., 2017; Kober et al., 2013), successes in real-world business problems such as in supply chain optimization have been more elusive. Specifically, we consider the periodic review inventory control problem where a decision maker must decide how many units of inventory to stock to balance the cost of meeting customer demand with the cost of holding too much inventory. In the classical formulation, demand is a stochastic, non-stationary random variable, and there is a lead time between an order's placement and arrival. Under these dynamics, Madeka et al. (2022) is one of the first works that apply deep RL succesfully to a real-world supply chain problem.

A general requirement for applying RL to most real-world problems is a high-fidelity *simulator* under which an RL agent can explore new policies, as real-world exploration becomes prohibitively expensive and time-consuming. While simulation of game-play and robotics is well-understood due to simple or known dynamics, building a simulator for the inventory control problem is much more difficult due to unknown variability (e.g., in demand, lead time, etc) which makes counterfactual estimation difficult. Madeka et al. (2022) address this problem by using historical data to build a differentiable simulator that can be used to backtest *any* policy. Specifically, by formulating the problem as an exogenous decision process (Sinclair et al., 2022), in which most of the inputs are not affected by the agent, Madeka et al. (2022) prove learnability results for the inventory control problem.

In this work, we seek to relax the strict assumption on vendor lead time made by the classical inventory control model and instead use a learned model of transition dynamics. Vendor lead time is a critical component of inventory control, and has been studied in many previous works (Zipkin, 1986; Svoronos and Zipkin, 1991), under the aforementioned assumption that the entire order necessarily arrives all at once. However, in reality, vendors typically send inventory purchase orders out in multiple shipments and can send more or less than the requested amount — or even nothing at all. In this work we make the milder assumption that the vendor implicitly fulfills orders depending on an unknown, but exogenous factors such as supply and desired fulfillment rate.

Our proposed dynamics model is a generative deep learning model, Generative-Quantity over Time (Gen-QOT), capable of estimating the full joint probability distribution of inventory arrivals over time, that can be used within a historical simulator to accurate train and backtest RL policies. This work builds on much of the current progress in generative modeling for natural language processing by learning latent hierarchical representations of historical time-series like orders and arrivals while predicting a sequence of tokens corresponding to arrivals. Gen-QOT can be used to capture uncertainty in vendor lead times, splits into multiple shipments, non-unit arrival rates, and also the transformations and post-processors an order may be subject to in a typical supply chain such as carton rounding, vendor selection, and SKU aggregation. We show that accurate modeling of the arrival dynamics is sufficient for learning and backtesting any policy. Using data from a real-world A/B test of an RL agent learned on this simulator, we validate that our model is accurate *off-policy* (with respect to the data collection policy) which is critical for the backtest in the simulator to be valid. Finally, we provide empirical evidence that our Gen-QOT arrivals model is accurate, calibrated, and RL agents trained under our arrivals model outperforms those trained under baseline lead time models.

**Main contributions and organization**  Our main contribution are the following: (1) we formulate the inventory control problem with general arrivals as an exogenous decision process, and show that it is efficiently learnable from historical data by reduction to Theorem 3 of Madeka et al. (2022); (2) we introduce a novel generative model of arrivals, (3) we demonstrate by simulation studies and a real world evaluation the importance of an accurate model of the arrival dynamics, (4) using data from a real-world A/B test we show that our learned dynamics model remains accurate on *off-policy* states induced by the RL agent's behavior in the actual supply chain. The remainder of this paper is organized as follows: In Section 2 we provide a brief overview of related work. Then, in Section Section 3 we present our inventory control problem formulated as an Interactive Decision Process (IDP), under our new setting for inventory arrivals. Finally, in Section 4 we showcase some empirical results to support our claims.

## 2  RELATED WORK

### 2.1  RL FOR INVENTORY CONTROL

Many papers in the literature have studied how to efficiently train RL agents, but notably, it is difficult to execute exploration policies in the real-world without suffering large losses. A common practical workaround is to take a *Sim2Real RL* approach, by building a simulator under which to train an RL policy. However, even with a sufficiently large amount of offline or simulated data, efficient algorithms for RL are generally intractable (Wang et al., 2020; Du et al., 2019; Shariff and Szepesvári, 2020; Weisz et al., 2021). There has been recent optimism that an RL-based solution for the inventory control problem can outperform existing baseline methods, due to the observation that much of the variability within the inventory control problem appears to be exogenouse of the decision-making policy (Sinclair et al., 2022). Specifically, Madeka et al. (2022) build a differentiable simulator (see Suh et al. (2022)) using historical inventory and sales data, and successfully train and backtest an RL agent to achieve improved real world performance over traditional methods. Furthermore, they present theoretical results that demonstrate a novel reduction from reinforcement learning to supervised learning, when the stochastic aspects (i.e., demand, lead time, costs, and selling price) of the system are exogenous. We also note for interested readers that deep RL approaches for the inventory control problem have also been studied in Gijsbrechts et al. (2022); Qi et al. (2023). We note that in all of the prior works using RL to solve the inventory control problem, vendor lead time is the assumed model for inventory arrivals.

## 2.2 Deep Generative Modeling

A number of deep generative techniques have been developed to estimate the likelihood of observations in training data and generate new samples from the underlying data distribution. For example, generative adversarial networks (Goodfellow et al., 2014) simultaneously train a generator to produce realistic samples and a discriminator to classify samples as either real or generated. Variational Autoencoders (Kingma and Welling, 2013; 2019) try to find a lower dimensional representations of the data by assuming a parametric posterior for some latent variable and minimizing the Kullback–Leibler divergence between the true and approximated posterior of this latent variable. Finally, auto-regressive models estimate the conditional distribution of the current element in a sequence given the previous elements and have been used successfully in image generation (Van Oord et al., 2016), NLP (Brown et al., 2020; Devlin et al., 2019), and time-series forecasting (Madeka et al., 2018). Our work employs autoregressive modeling by decomposing the full problem of estimating the joint distribution of arrivals into the simpler problem of merely predicting the next arrival in a sequence given the previously realized shipments.

## 3 Inventory Control Problem with Quantity over Time Arrivals

In this section, we will follow the Interactive Decision Process (IDP) formulation of Madeka et al. (2022), borrowing most of the conventions and notation, except we define and treat the "lead time" process differently. At a high level, a central planner is trying to determine how many units of inventory to order at every time step $t = 1, 2, ..., T$, in order to satisfy demands $D_t$. The goal is to maximize profits by balancing having enough inventory on hand to satisfy demand (we assume a lost sales customer model), with the cost of holding too much inventory. The dynamics we are most concerned with is the order arrivals process. The standard formulation in the literature is the vendor lead time (VLT) arrivals model, where upon placing an inventory order decision $a_t$ at time $t$, a single quantity $v_t$ is drawn from an exogenous lead time distribution, and the entire order arrives $v_t$ time steps later at time $t + v_t$.

Instead, we propose a novel quantity over time (QOT) arrivals model. In the QOT arrivals model, we assume that orders can arrive in multiple shipments over time, and the total arriving quantity may not necessarily sum up to the order quantity placed. At every time $t$, the vendor has allocated a supply $U_t$ that denotes the maximum number of units it can send (regardless the amount we order), which will arrive over from the current week up to $L$ weeks in the future according to an exogenous arrival rate vector $(\rho_{t,0}, ..., \rho_{t,L})$. That is, the arrivals at lead time $j$ from order $a_t$ is equal to $\min(U_t, a_t)\rho_{t,j}$. We denote the arrival quantity as $o_{t,j} := \min(U_t, a_t)\rho_{t,j}$. Here, we implicitly assume that orders necessarily arrive after $L$ time steps, and critically we do not assume that $\sum_{j=0}^{L} \rho_{t,j}$ is necessarily 1.

### 3.1 Mathematical notation

Denote by $\mathbb{R}$, $\mathbb{R}_{\geq 0}$, $\mathbb{Z}$, and $\mathbb{Z}_{\geq 0}$ the set of reals, non-negative reals, integers, and non-negative integers, respectively. Let $[ \cdot ]$ refer to the set of positive integers up to the argument, ie. $[ \cdot ] = \{x \in \mathbb{Z} \mid 1 \leq x \leq \cdot \}$. The inventory management problem seeks to find the optimal inventory level for each product $i$ in the set of retailer's products, which we denote by $\mathcal{A}$. We assume our exogenous random variables are defined on a canonical probability space $(\Omega, \mathcal{F}, \mathbb{P})$, and policies are parameterized by $\theta$ in some parameter set $\Theta$. We use $\mathbb{E}^{\mathbb{P}}$ to denote an expectation operator of a random variable with respect to some probability measure $\mathbb{P}$. Let $||X, Y||_{TV}$ denote the total variation distance between two probability measures $X$ and $Y$.

### 3.2 IDP Construction

Our IDP is governed by external (exogenous) processes, a control process, inventory evolution dynamics dynamics, and a reward function. To succinctly describe our process, we focus on just one product $i \in \mathcal{A}$, though we note our decisions are made jointly for every product.

*External Processes*: At every time step $t$, for product $i$, we assume that there is a random demand process $D_t^i \in [0, \infty)$ that corresponds to customer demand during time $t$ for product $i$. We also assume that the random variables $p_t^i \in [0, \infty)$ and $c_t^i \in [0, \infty)$ are the random variables corresponding to selling price and purchase cost. The supply $U_t^i \in [0, \infty)$ corresponds to the maximum amount of

inventory the vendor is allowed to send. Finally, the arrival rate processes $\boldsymbol{\rho}_t^i := \{\rho_{t,j}^i\}_{j=0}^L$ describe the arrivals over the next $L$ time steps from an order placed at the current time $t$. Our exogenous state vector for product $i$ at time $t$ is all of this information:

$$s_t^i = (D_t^i, p_t^i, c_t^i, U_t^i, \boldsymbol{\rho}_t^i).$$

In practice, we will empirically tackle the problem where there is a joint distribution of the processes over all products $i$, though for our learnability results in Section 3.3 we will assume independence of the processes between products. Therefore, we will define the history

$$H_t := \{(s_1^i, ..., s_{t-1}^i)\}_{i=1}^{|\mathcal{A}|}$$

as the joint history vector of the external processes for all the products up to time $t$.

*Control Processes*: Our control process will involve picking actions for each product jointly from a set of all possible actions $\mathbb{A} := \mathbb{R}_{\geq 0}^{|\mathcal{A}|}$. For product $i$, the action taken is denoted by $a_t^i \in \mathbb{R}_{\geq 0}$, the order quantity for product $i$. For a class of policies parameterized by $\theta$, we can define the actions as

$$a_t^i = \pi_{\theta,t}^i(H_t).$$

We characterize the set of these policies as $\Pi = \{\pi_{\theta,t}^i | \theta \in \Theta, i \in \mathcal{A}, t \in [0, T]\}$.

*Inventory Evolution Dynamics*: We assume that the implicit endogenous inventory state follows standard inventory dynamics and conventions. In particular, we assume that ordered inventory arrives at the beginning of the time period, so the inventory state transition function is equal to the order arrivals at the beginning of the week minus the demand fulfilled over the course of the week. We note that both demand and arrivals may be censored due to having lower inventory on-hand or vendor having low supply, respectively. The amount arriving, according to our model of arrivals is:

$$I_{t_-}^i = I_{t-1}^i + \sum_{j=0}^L \min(U_{t-j}^i, a_{t-j}^i)\rho_{t-j,j}^i, \tag{3.1}$$

where $I_t^i$ is the inventory at the end of time $t$, and $I_{t_-}^i$ is the inventory at the beginning of time $t$, after arrivals but before demand is fulfilled. Then, at the end of time $t$, the inventory position is:

$$I_t^i = \min(I_{t_-}^i - D_t^i, 0).$$

*Reward Function*: The reward at time $t$ for product $i$ is defined as the selling price times the total fulfilled demand, less the total cost associated with any newly ordered inventory (that will be charged by the vendor upon delivery):

$$R_t^i = p_t^i \min(D_t^i, I_{t_-}^i) - c_t^i \min(U_t^i, a_t^i)(\sum_{j=0}^L \rho_{t,j}^i). \tag{3.2}$$

We will write $R_t(H_t, \theta)$ to emphasize that the reward is a function only of the exogenous $H_t$ and the policy parameters $\theta$. Recall that selling price and buying cost are determined exogenously. We assume all rewards $R_t^i \in [R^{min}, R^{max}]$, and assume a multiplicative discount factor of $\gamma \in [0, 1]$ representing the opportunity cost of reward to the business. Again, we make the dependence on the policy explicit by writing $R_t^i(\theta)$. The objective is to select the best policy (i.e., best $\theta \in \Theta$) to maximize the total discounted reward across all products, expressed as the following optimization problem:

$$\max_\theta \mathbb{E}_\mathbb{P}[\sum_{i \in \mathcal{A}} \sum_{t \in [0,T]} \gamma^t R_t^i(\theta)] \tag{3.3}$$

subject to:

$$I_0^i = k^i$$

$$a_t^i = \pi_{\theta,t}^i(H_t)$$

$$I_{t_-}^i = I_{t-1}^i + \sum_{j=0}^L \min(U_{t-j}^i, a_{t-j}^i)\rho_{t,j}^i, \tag{3.4}$$

$$I_t^i = \min(I_{t_-}^i - D_t^i, 0).$$

Here, $\mathbb{P}$ denotes the joint distribution over the exogenous processes. The inventory $I_0^i$ is initialized at $k_i$, a known quantity *a priori*.

### 3.3 Simulator Construction, Learning, and Learnability

#### Learning Objective

For the policy to be efficiently learnable, we need to restrict the policy for product $i$ at time $t$ to be a function only of the history of item $i$, $H_t^i := \{(s_1^i, ..., s_{t-1}^i)\}$, and the learnable parameter $\theta$ is shared by all item's policies. The reward is therefore now a function $R_t(H_t^i, \theta)$ of only the history of item $i$ and the parameter $\theta$. The learning objective then becomes

$$J_T(\theta) := \mathbb{E}[\sum_{i \in \mathcal{A}} \sum_{t \in [0,T]} \gamma^t R_t^i(\theta)],$$

which we estimate via simulation with the objective

$$\widehat{J}_T(\theta) := \sum_{i \in \mathcal{A}} \sum_{t \in [0,T]} \gamma^t R_t^i(\theta).$$

This is clearly an unbiased estimate of $J_T$ as the historical data $H_T$ is exogenous of the choice of policy. Note that as in Madeka et al. (2022) the objective and dynamics are differentiable, and thus we can use a DirectBackprop style algorithm.

#### Learnability

Our problem formulation fits under the framework described in Madeka et al. (2022), with additional exogenous variables $U_t$ and $\boldsymbol{\rho}_t$ as part of the external state process. Hence, assuming full observability of these processes, we would be able to learn and backtest efficiently because we could accurately simulate the value of any policy. This follows immediately from Theorem 2 of Madeka et al. (2022) as we assume that the supply and fill rate processes are exogenous.

In reality, one does not fully observe the supply of the vendors ($U_t^i$) or the fill rates ($\boldsymbol{\rho}_t^i$), so we require an accurate forecast of these processes. We will use additional observed exogenous context to forecast these unobserved components $x_t^i \in \mathbb{R}^D$ that is available at time $t$ for product $i$. The context history is denoted as $X_T^i := (x_1^i, \dots, x_T^i)$.

**Assumption 3.1** (Accurate Forecast of Supply and Fill Rates). Let $H_{T,F}^i := (U_1^i, \boldsymbol{\rho}_1^i, \dots, U_T^i, \boldsymbol{\rho}_T^i)$ denote the history of the unobserved exogenous supply and fill processes through time $T$. Likewise denote the observed components of the exogenous history $H_T^i$ as $H_{T,O}^i$. Now, we can consider the distributions $\mathbb{P}_F^i := \mathbb{P}(H_{T,F}^i | H_{T,O}^i, X_T^i)$ and $\widehat{\mathbb{P}}_F^i := \widehat{\mathbb{P}}(H_{T,F}^i | H_{T,O}^i, X_T^i)$. If

$$\frac{1}{|\mathcal{A}|} \sum_{i \in \mathcal{A}} ||\widehat{\mathbb{P}}_F^i, \mathbb{P}_F^i||_{TV} \le \epsilon_F,$$

we call $\widehat{\mathbb{P}}_F^i$ an accurate forecast of $\mathbb{P}_F^i$.

Under Assumption 3.1, it follows from Theorem 3 of Madeka et al. (2022) that the inventory control problem with general arrivals is efficiently learnable in the case where we do not observe the supply and fill rate processes. In practice, we may choose to forecast *arrivals* instead of the supply and fill rate processes (see Remark 3.2).

**Remark 3.2** (Forecasting Arrivals). Note that the dynamics (3.4) and reward function (3.2) depend only on the arrivals $o_{t,j}^i := \min(U_t^i, a_t^i)\rho_{t,j}^i$, so we forecast arrivals conditional on the action $a_t^i$ rather than the supply and fill processes.

## 4 Gen-QOT and Empirical Results

Having established that our problem of interest is efficiently learnable, we proceed with describing the QOT model and then evaluating the model In this section we first describe, at a high-level, what we model with Gen-QOT. We then validate that the model by backtesting performance using historic

data, and proposing several calibration metrics for evaluating the quality of sample paths. We then demonstrate (in simulation studies) that having an accurate model of arrivals is important for policy learning. Finally, we present accuracy metrics on off-policy data (relative to the policy that generated the training data) gathered during a real world A/B test of an RL inventory control policy trained in the simulator described in the previous section.

## 4.1 THE GEN-QOT MODEL

Per remark 3.2, we forecast the arrival sequence directly rather than the supply and arrival processes. Formally, we forecast the distribution

$$p(o_{t,0}^i, \ldots, o_{t,L}^i | H_t^i, X_t^i, a_t).$$

The model to minimize log-likelihood of the forecasted distribution. See Appendix B for a complete description of the model and training objective.

## 4.2 TRAINING AND EVALUATION DATA

We train Gen-QOT on inventory orders from 250K products from the US marketplace from 2017-05-13 to 2019-03-23 and holdout 100K actions from 2017-05-13 to 2020-03-21 to evaluate model performance. This time period allows us to judge both in-time and out-of-time time generalization.

## 4.3 EVALUATING GEN-QOT ON HISTORIC DATA

Our backtest of the Gen-QOT model on historic data consists of two components: (1) we propose several calibration metrics that measure important properties of generated sample paths, (2) we backtest the Gen-QOT model out-of-sample against a baseline direct quantile forecast.

In Appendix A we perform a more qualitative analysis of the generated sample paths versus real sample paths, and in Appendix E we perform an ablation study across different candidate architectures for Gen-QOT.

### 4.3.1 CALIBRATION METRICS FOR SAMPLE PATHS

Designing metrics to measure the goodness-of-fit of sample paths drawn from the estimated joint distribution of inventory arrivals is non-obvious because drawing sample paths requires recursively sampling model outputs. Log-likelihood can be used to confirm the quality of the first predicted arrival, but to estimate the likelihood of subsequent predictions, we would need access to the actual next arrival of the sampled trailing sequence. This problem mirrors the well-known task of evaluating sequences in NLP, where researchers rely on heuristics like the Bilingual Evaluation Understudy (BLEU) score that are known to correlate with human judgements of quality but have no theoretical guarantee of goodness.

Despite being trained on log-likelihood, it is not clear whether Gen-QOT captures important features of the inventory arrival process that are necessary to accurately simulate the behavior or an inventory control policy. Instead, we can draw on expert knowledge and propose several several calibration metrics to establish the quality of the predicted inventory arrival distribution:

1. Does Gen-QOT predict the right amount of cumulative inventory $l$ weeks after an order is placed?
2. Do Gen-QOT predict receiving zero inventory for the correct orders?
3. Do Gen-QOT predict correctly whether there is an arrival in the first week after an order is placed?

To verify the first objective, the calibration of cumulative arrivals, we regress the actual cumulative fill rate on the mean predicted cumulative fill rate for each week, $i$. To verify the second and third objectives we use classifier calibration plots.

Figure 1 shows this calibration plot for in-time samples of actual vs. predicted mean for normalized cumulative fill rate at the end of the fourth week. In this case the regression coefficient is 1.0295

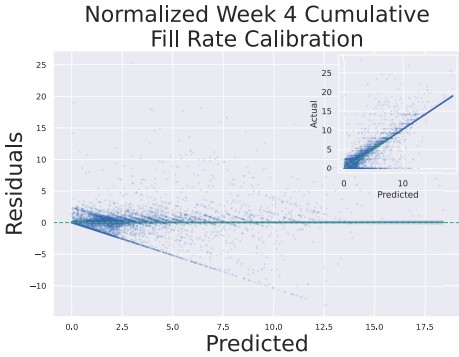

Figure 1: Residual calibration plot for cumulative fill rate – residuals are plotted against predicted values in the main figure, while the original plot is shown in an inner figure. All numbers are normalized by actual average cumulative fill rate

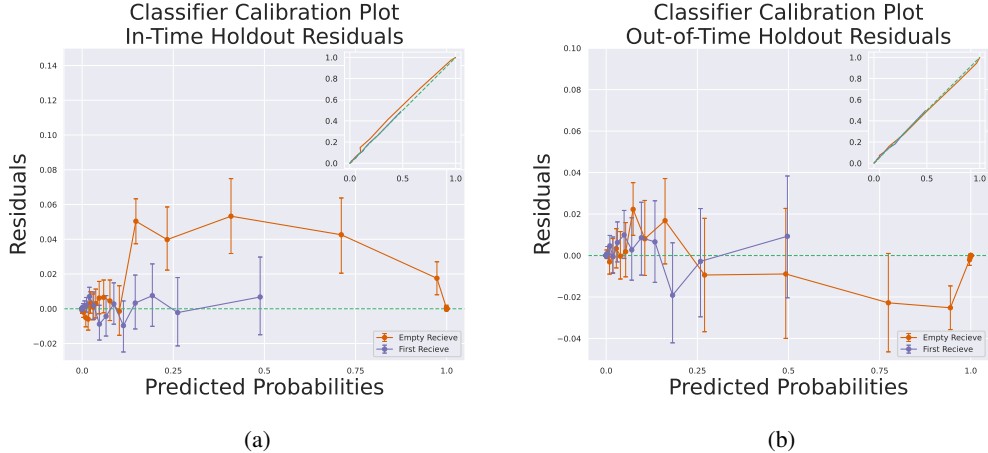

Figure 2: Residual calibration plots for Gen-QOT – residuals are plotted against predicted values in the main figure, while the original plot is shown in an inner figure.

with 95% confidence interval of $[1.020, 1.039]$. A full set of regression coefficients for in-time and out-of-time evaluation across all weeks is included in Appendix F. While the confidence interval is slightly higher than ideal, it still suggests that Gen-QOT roughly predicts the correct cumulative quantity of inventory arrivals.

The classifier calibration plots to validate the predicted probabilities of an empty arrival sequence and arrival in the first week are produced by discretizing and binning the predicted probabilities and then estimating the mean of the actual classifications for each bin. Ideally, the average actual will be equal to the mean of the predicted probabilities in each bin, and this point will fall along the 45° line.

Figure 2a and Figure 2b show the calibration of Gen-QOT at predicting whether a purchasing action will yield zero inventory for both the in-time holdout and out-of-time holdout. In both cases, points generally follow the ideal calibration line, with some slight and expected degradation in the out-of-time holdout. These figures also depict the calibration of Gen-QOT at predicting whether inventory will arrive in the first week after an action for both the in-time holdout and out-of-time holdout. Like the calibration check for whether an action returned zero inventory, points generally fall along the 45° line indicating Gen-QOT as a classifier for receiving inventory in the first week after an action's execution is reasonably well calibrated.

### 4.3.2 COMPARING GENERATIVE TO DIRECT FORECASTING FOR VENDOR LEAD TIMES

Because many inventory control systems rely on simplified optimization models that assume only random lead-time and ignore quantity stochasticity, as a baseline, we evaluate the Gen-QOT model against a variant model architecture that directly predicts quantiles of the vendor lead-time distribution. To directly predict quantiles, we replace the recurrent neural-net decoder with a simple multi-layer perceptron. We train both models with the same features on the product set described in Section 4.2 but use 2016-17 as a training period, 2018 as a validation period, and 2019 as a test period.

To estimate vendor lead-time using Gen-QOT, which estimates the distribution of sequences of receives rather than a single lead-time, we generate 32 sample paths and compute the empirical quantiles across all receives in the sampled sequences of receives. The results are summarized in Table 1 where we compare both models using a quantity weighted quantile loss. We find that on the test period, that while directly predicting vendor-lead times leads to a lower quantile loss, the estimate produced via samples from Gen-QOT is competitive.

Table 1: Quantile loss of generative model versus direct quantile forecast.

| Model | CRPS | p10 | p30 | p50 | p70 | p90 |
|---|---|---|---|---|---|---|
| Direct Quantile Prediction | **100.00** | 100.00 | 100.00 | **100.00** | **100.00** | **100.00** |
| Gen-QOT | 101.60 | **92.58** | **97.76** | 100.01 | 104.52 | 112.53 |

### 4.4 SIMULATION AND REAL-WORLD EXPERIMENTS

To validate that the more realistic inventory arrival dynamics predicted by Gen-QOT are useful for training inventory control policies, we compare two RL policies trained using Direct Backprop (Madeka et al., 2022) policies, one trained under transition dynamics dictated by Gen-QOT and another trained using a baseline lead-time model (Januschowski et al., 2022). Evaluation is performed on a held-out test-period under transition dynamics governed by Gen-QOT.

Table 2 summarizes the changes in the sum of discounted reward of both policies relative to a baseline policy. We find that the direct back-prop inventory control policy trained with transition dynamics given by Gen-QOT outperforms the policy trained using the baseline lead-time model when evaluated under Gen-QOT transition dynamics. While it is unsurprising that the policy trained on the simulator (with Gen-QOT) used in evaluation performs best, this does underscore the importance of using a simulator for policy learning that has an accurate model of arrival dynamics (in this case, the gain is 8%).

Table 2: Comparison of RL policies trained with Gen-QOT transitions versus transitions produced by a baseline lead-time model.

| Policy | Discounted Reward |
|---|---|
| News-vendor Baseline | 100.00% |
| Vendor Lead-Time DirectBP | 109.64% |
| Gen-QOT DirectBP | **117.81**% |

### 4.5 OUT OF SAMPLE (AND OFF-POLICY) PERFORMANCE ON REAL WORLD DATA

Finally using data generated via a randomized control trial comparing an inventory control policy trained with Gen-QOT to a newsvendor style policy (the policy used to collect the training data for Gen-QOT), we validate that the errors under the treatment arm are statistically indistinguishable from errors under the control policy. This is critical because in order for our *inventory control backtest* to be accurate, we required Assumption 3.1. In Table 3 we see that the difference in forecast performance on-policy versus off-policy in the actual supply chain is not statistically significant.

Table 3: Difference in quantile loss of the distributions forecasted by Gen-QOT across the control (on-policy) and treatment (off-policy) arms of a real-world A/B test of an inventory control policy learned from a simulator using Gen-QOT.

| Quantile | Mean Quantile Loss Difference | [0.025 | 0.075] |
|----------|-------------------------------|--------|--------|
| p10 | -0.02% | -0.06% | 0.02% |
| p30 | -0.01% | -0.06% | 0.03% |
| p50 | -0.01% | -0.07% | 0.04% |
| p70 | -0.01% | -0.08% | 0.04% |
| p90 | -0.01% | -0.09% | 0.06% |
| p98 | 0.00% | -0.11% | 0.09% |

## 5 CONCLUSION

We presented a new model of inventory control, by generalizing the standard lead time assumption that is typically made in the literature. Instead, we assume a more natural inventory arrival process, allowing orders to arrive, e.g. over time in multiple shipment, not all at once, and not necessarily the exact order quantity. Our arrivals dynamics exhibit desirable properties that allow a differentiable simulator to be built with historical data. This simulator can be used to efficiently learn and backtest reinforcement learning policies, provided we can accurately model the arrival dynamics following an order. Via backtesting against out of sample data and via simulation studies, we showed that Gen-QOT is calibrated and indeed results in better inventory control policies. Finally, by evaluating the model on real world trajectories collected from an A/B test, we saw that even when off-policy, Gen-QOT's forecasted distributions were no less accurate than on-policy trajectories.

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

## A  QUALITATIVE ANALYSIS OF GENERATED ARRIVAL SEQUENCES

An example set of order-quantity normalized samples drawn from the estimated joint probability distribution of inventory arrivals over time is shown in Figure 3 and contrasted with set of real order-quantity normalized inventory arrivals. We find that these simulated paths reflect several important properties of realistic inventory arrivals including discontinuity over time, non-zero probability for receiving zero inventory, and possibility of receiving more and less inventory than the requested.

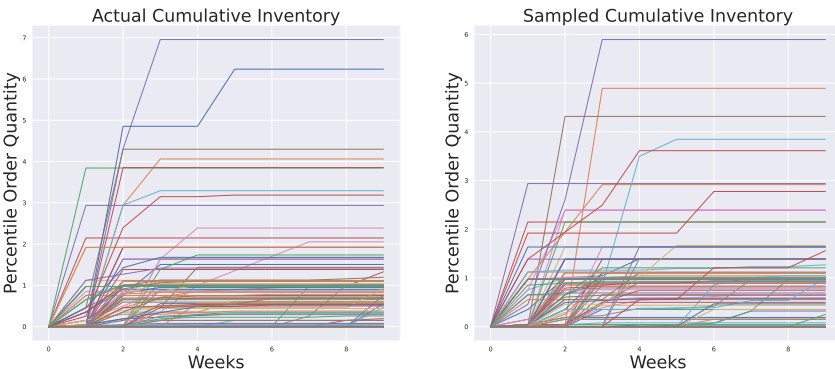

Figure 3: A set of 256 real and simulated sample paths of order-quantity normalized inventory arrivals.

## B  THE GEN-QOT MODEL

In this section, we describe our novel arrivals prediction model. First, denote historical covariates $x_t^i$ for each product $i$ at each time $t$ that will be used to estimate the distribution of state transition probabilities. The vector $x_t^i$ can be thought of as the observational historical data that contains at minimum, information such as the time-series of historical orders and arrivals. Other information that can be incorporated includes vendor and product attributes, existing geographic inventory allocation, and the distance to various holidays that may affect the ability for vendors and logistics providers to reliably fulfill and ship inventory. Also denote the actual arrivals of inventory as $\{o_{t,j}^i\}_{j=0}^L$, where $o_{t,j}^i \in \mathbb{R}_{\geq 0}$. Recall $L$ is the maximum possible lead time for an arrival.

The Gen-QOT model solves the problem of predicting the joint distribution of inventory arrivals for an action $a_t^i$ at time $t$. To model the distribution of arrival sequences $\{o_{t,j}^i\}_{j=1}^L$, we consider the distribution over arrival rates[1] instead:

$$\alpha_{t,j}^i := \begin{cases} \frac{o_{t,j}^i}{a_t^i} & a_t^i \neq 0 \\ 0 & a_t^i = 0. \end{cases}$$

Our goal then is to produce a generative model from which we can sample sequences of arrival rates $\{\alpha_{t,j}^i\}_{j=1}^L$. Note that these arrival rates *do not* need to sum to 1.

Next, observe that an equivalent formulation is to model a sequence of tuples of arrival rates and time since the last non-zero arrival: $\{(k_{t,s}^i, \widetilde{\alpha}_{t,s}^i)\}_{s \in \mathbb{Z}_{\geq 0}}$, where $\widetilde{\alpha}_{t,s}^i$ denotes the proportion of $a_t^i$ in the $s^{th}$ arrival and $k_{t,s}^i$ denotes the number of periods since the previous non-zero arrival. By convention the first arrival is measured as an offset from $t - 1$.

### B.1  PROBABILISTIC MODEL

At a high-level, the methodology we employ to predict a sequence of arrivals is to construct a grid over quantity and time that can be used to bin individual arrivals into distinct arrival classes. Our

---

[1]Note that these rates are slightly different than $\rho$ in the previous section as this one does not consider a supply variable.

model then produce a categorical distribution over these classes conditioned on previous arrivals in the sequence, akin to generative sequence modeling in NLP. Once the sequence of classes has been sampled, we can map it back to the original quantity of interest using the function $V$, defined in Appendix B.1.2.

### B.1.1 MODEL THE DISTRIBUTION BY BINNING

We rely on binning to avoid making parametric assumptions about the distributions of arrivals. To model the proportion of requested inventory in each arrival and number weeks since last arrival, we can bin these pairs into classes. We define a sequence of $N$ grid points $\tau^{(1)}, \ldots, \tau^{(N)}$ over time periods since last arrival and $M$ grid points $p^{(1)}, \ldots, p^{(M)}$ for the proportions, such that

$$\tau^{(n)} \leq \tau^{(m)} \ \ \forall n < m,$$
$$p^{(n)} \leq p^{(m)} \ \ \forall n < m,$$
$$\tau^{(1)} = p^{(1)} = 0,$$
$$\tau^{(N)} = \tau_{max}$$
$$p^{(M)} = p_{max}$$

where $\tau_{max}$ and $p_{max}$ are both large constants. For each $n, m \in [N-1] \times [M-1]$, define the set

$$Z_{n,m} := \{(k, \alpha) : \tau^{(n)} \leq k < \tau^{(n+1)} \text{ and } p^{(m)} \leq \alpha < p^{(m+1)}\}.$$

By construction, $\mathcal{Z} := \{Z_{n,m} : (n, m) \in [N-1] \times [M-1]\}$ is a partition of $\{u \in \mathbb{Z} \mid 0 \leq u < \tau_{max}\} \times \{v \in \mathbb{R} \mid 0 \leq v < p_{max}\}$, the space of all possible pairs $(k_s^i, \widetilde{\alpha}_{t,s}^i)$.

Next we denote the index pair $(n, m)$ that corresponds to a specific arrival class $Z_{n,m}$ by the random vector $\zeta$. For example $\zeta = (1, 2)$ is a reference to the arrival class $Z_{1,2}$. Additionally, we add a special index pair denoting the end of an arrivals sequence, which we can signify with $\emptyset$. This $\emptyset$ can be thought of as a special arrival that designates that the sequence of arrivals from an action has terminated. In practice, we can use $(0, 0)$ to represent the value of $\emptyset$. Given this construction, we have $\zeta$ taking values in $\widetilde{\mathcal{Z}} := \{[N-1] \times [M-1]) \cup \{(0, 0)\}\}$

Our objective is now to estimate the joint probability of a sequence of index pairs $\zeta_{t,0}^i, \zeta_{t,1}^i \ldots$ corresponding to a sequence of arrival classes given some $a_t^i$. Gen-QOT specifically estimates

$$P(\zeta_{t,0}^i, \zeta_{t,1}^i, \ldots | x_t^i, a_{t,0}^i) = \prod_j P(\zeta_{t,j}^i | \zeta_{t,j-1}^i, \zeta_{t,j-2}^i, \ldots, \zeta_{t,0}^i, x_t^i, a_t^i) \quad \text{(B.1)}$$

and decomposes this joint probability into the product of conditional probabilities.

### B.1.2 GENERATING SAMPLES OF INVENTORY ARRIVALS: MAPPING FROM CLASSES TO ACTUAL ARRIVALS

Sampling this joint probability distribution gives a sequence of arrival classes $(\zeta_{t,0}^i, \zeta_{t,1}^i, \ldots)$. To obtain a sample path of inventory arrivals for an action $a_t^i$, we use the function $V : \widetilde{\mathcal{Z}} \to \mathbb{R}_{>0} \times \mathbb{Z}_{>0}$ to map each element $\zeta_{t,s}^i = (n, m)$ of the sampled sequences of arrival classes back to a tuple of arrival arrival rates and time-since-last-arrival. This function is implemented by mapping each of the tuples $(n, m)$ to a representative element $(\bar{k}_{n,m}, \bar{\alpha}_{n,m})$ in the corresponding $Z_{i,j}$. Here, $\bar{k}_{n,m}$ and $\bar{\alpha}_{n,m}$ denote the mean of the respective quantities observed in the data in each arrival class bin, for every $n, m$. However, we note that other quantities can be used, e.g. the center of the bin of each arrival class $(\frac{\tau^{(n)} + \tau^{(n+1)}}{2}, \frac{p^{(i)} + p^{(i+1)}}{2})$.

The sampled sequence of arrival class indexes, $(\zeta_{t,0}^i, \zeta_{t,1}^i, \ldots)$, can be transformed back into the original quantity of interest $o_{t,j}^i$ by substituting each $\zeta_{t,s}^i$ with the tuple $(\bar{k}_{n,m}, \bar{\alpha}_{n,m})$ to recover a sequence of estimated periods since last arrival and proportion of inventory, $(k_{t,s}^i, \widetilde{\alpha}_{t,s}^i)$. By cumulatively summing the periods, and multiplying each $\widetilde{\alpha}_{t,s}^i$ by $a_t^i$, we immediately recover every $o_{t,j}^i$.

### B.1.3 EXAMPLE OF ARRIVAL SEQUENCE TRANSFORMATION

**Inventory arrivals to arrival class sequence:** As an example, let $a_t^i = 10$, and the true arrival sequence be $\langle 0, 3, 5, 0, 4 \rangle$. Additionally we can construct a series of grid-points $\tau^{(l)} = l - 1$ and $p^{(m)} = 0.2 \cdot (m - 1)$ for all $l \in [L]$ and $m \in [M]$ where $L = 4$ and $M = 6$.

Then we can map from the original sequence of arrivals over time $\{o_{t,j}^i\}$ to tuples of arrival rates and time since last arrival $\{(k_s^i, \widetilde{\alpha}_{t,s}^i)\}$ by normalizing by action and computing the time since last arrival.

$$\langle 0, 3, 5, 0, 4 \rangle \;\longrightarrow\; \langle (2, 0.3), (1, 0.5), (2, 0.4) \rangle$$

Using the grid-points and $(0, 0)$ as a reference for $\zeta$, we can transform the sequence of $\{(k_s^i, \widetilde{\alpha}_{t,s}^i)\}$ into a sequence of $\zeta$'s by binning the tuples of time since last arrival and inventory percentile. For example the tuple $(2, 0.3)$ is placed in the bin with coordinates $(2, 2)$ because it is 2 time periods since the start of the sample and $0.3$ falls in the second bin that sits between $0.2$ and $0.4$. The rest of the sequence is transformed as follow:

$$\langle (2, 0.3), (1, 0.5), (2, 0.4) \rangle \;\longrightarrow\; \langle (2, 2), (1, 3), (2, 3), (0, 0) \rangle$$

**Arrival class sequence to inventory arrivals:** We can reverse this example by imagining Gen-QOT generated the sequence $\langle (2, 2), (1, 3), (2, 3), (0, 0) \rangle$. We can replace each $Z_{n,m}$ with the representative element corresponding to the middle of the bin $(\bar{k}_{n,m}, \bar{\alpha}_{n,m}) = (l, 0.2 \cdot m - 0.1)$. Substituting into the sequence we get

$$\langle (2, 2), (1, 3), (2, 3), (0, 0) \rangle \;\longrightarrow\; \langle (2, 0.3), (1, 0.5), (2, 0.5) \rangle$$

Then we can cumulatively sum the time since last arrival and multiply by the action to recover sampled sequence of arrivals over time, $\{o_{t,j}^i\}$

$$\langle (2, 0.3), (1, 0.5), (2, 0.5) \rangle \;\longrightarrow\; \langle 0, 3, 5, 0, 5 \rangle$$

In this example we see two crucial features of the probabilistic model utilized by Gen-QOT. Firstly, the sum of arrival quantities over actual and sampled arrival quantities do not need to be equal to the action $a_t^i$. Secondly, the structure of the grid and choice of representative unit can induce error in the estimated sample if not carefully chosen.

## C  NEURAL ARCHITECTURE AND LOSS

Following canonical work in generative modeling for language (Sundermeyer et al., 2010; Graves, 2013), our work uses recurrent neural-networks to generate sequences of arrival classes. Additionally, following van den Oord et al. (2016); Wen et al. (2017) we rely on stacked and dilated temporal convolutions to learn a useful embedding of historical sequences of arrivals, order quantities, and OQs. Merging the architectures together, Gen-QOT is implemented using a encoder-decoder style architecture with a dilated temporal convolution encoder and recurrent decoder. Full model hyper-parameters along with arrival class definitions can be found in Appendix D. Additionally, a richer comparison of various model architectures can be found in Appendix E.

The network is optimized to maximize the likelihood of generated samples by being trained to minimize cross-entropy loss. Allowing $y$ to be a matrix of indicators across all arrival classes $Z_{n,m}$ and $\widehat{y}$ to be the matrix of probabilities for each arrival class, then the loss for a single prediction can be written as

$$J(y, \widehat{y}) = - \sum_{n=1}^{N-1} \sum_{m=1}^{M-1} y_{n,m} \log(\widehat{y}_{n,m})$$

where the sum runs over all arrival classes. Finally, the network is trained using the teacher-forcing algorithm (Williams and Zipser, 1989), where during training the model learns to predict the next token in a sequence given the actual trailing sub-sequence. During inference, strategies like beam-search (Graves, 2012) can be used to find the highest likelihood sequence of arrival classes. For our work, we implemented a simplified inference algorithm that samples the predicted distribution of arrival classes to generate a trailing sub-sequence that is used to predict subsequent token.

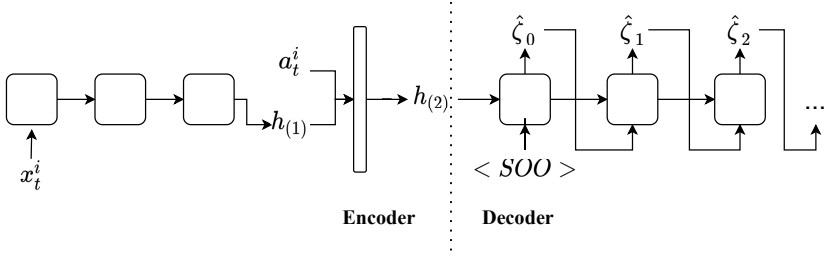

Figure 4: Visualization of Gen-QOT model architecture. The model structure uses a classic encoder decoder architecture with a dilated causal convolution encoder and standard recurrent decoder. The diagram above demonstrates how samples are generated the Gen-QOT model during inference, where $< SOO >$ is a vector of zeros

## D  GEN-QOT TRAINING HYPER-PARAMETERS

Table 4: Training hyper-parameters for Gen-QOT

| Hyper-parameter | Value |
|---|---|
| Epochs | 500 |
| Learning Rate | $1 \times 10^{-4}$ |
| Optimizer | Adam |
| Number of Convolution Layers | 5 |
| Number of Convolution Channels | 32 |
| Number of Recurrent Layers | 2 |
| Convolution Dilations | $[1, 2, 4, 8, 16]$ |
| Recurrent Decoder Size | 512 |
| Multi-layer Perception Size | 512 |
| Activation | ReLU |

## E  NEURAL ARCHITECTURE ABLATION

Given the broad set of neural architectures available for fitting *sequence-to-sequence* problems, we test a set of different encoder and decoder neural networks classes. Specifically we tested multi-layer perceptron vs causal-convolution encoder, and recurrent nueral network vs transformer decoder. In the end, we trained four models on data from sequences of arrivals from 10MM orders from 2017 and 2018 and tested on arrivals from 50K orders from 2019. To asses prediction quality, we on rely negative-log-likelihood of next token prediction, as well as unit weighted quantile-loss of cumulative quantity arrivals at 1, 4, and 9 weeks since UOQ computation execution.

Table 5: Results of ablation analysis for multiple metrics.

| Model | Metric | | | | |
|---|---|---|---|---|---|
| | QL of Cumulative Quantity of Arrivals: Week1 | | | | |
| | P10 | P30 | P50 | P70 | P90 |
| MLP-RNN | 100.00 | 100.00 | 100.00 | 100.00 | **100.00** |
| CNN-RNN | 70.15 | 98.60 | **89.15** | **96.67** | 100.42 |
| CNN-Transformer | **68.54** | **95.79** | 91.76 | 98.81 | 110.10 |
| | QL of Cumulative Quantity of Arrivals: Week4 | | | | |
| | P10 | P30 | P50 | P70 | P90 |
| MLP-RNN | 100.00 | 100.00 | 100.00 | 100.00 | 100.00 |
| CNN-RNN | 89.43 | 95.90 | 97.48 | 100.00 | 104.94 |
| CNN-Transformer | **29.27** | **39.85** | **52.04** | **66.75** | **80.86** |
| | QL of Cumulative Quantity of Arrivals: Week9 | | | | |
| | P10 | P30 | P50 | P70 | P90 |
| MLP-RNN | 100.00 | 100.00 | 100.00 | 100.00 | **100.00** |
| CNN-RNN | 100.79 | 100.58 | 100.82 | 101.70 | 102.01 |
| CNN-Transformer | **90.24** | **92.24** | **96.29** | **99.72** | 104.03 |
| | Negative Log-Likelihood of Next Token Prediction | | | | |
| MLP-RNN | 100.00 | | | | |
| CNN-RNN | **93.76** | | | | |
| CNN-Transformer | 94.07 | | | | |

## F  GEN-QOT CALIBRATION

Table 6: Calibration of cumulative inventory fill rate predicted $k$ weeks after submitting a order and actuals. Most coefficients are close to one, implying that Gen-QOT is able to accurately predict the cumulative quantity of inventory received for a specific order, $k$ weeks out. This calibration is important for a simulator to ensure that Gen-QOT predicts the correct quantity that has arrived.

| | In Time Holdout | | | Out of Time Holdout | | |
|---|---|---|---|---|---|---|
| Weeks After UOQ | Coefficient | [0.025 | 0.975] | Coefficient | [0.025 | 0.975] |
| 1 | 1.0577 | 1.056 | 1.06 | 1.0559 | 1.055 | 1.057 |
| 2 | 1.1393 | 1.138 | 1.141 | 1.1529 | 1.151 | 1.154 |
| 3 | 1.108 | 1.107 | 1.109 | 1.1311 | 1.13 | 1.132 |
| 4 | 1.0866 | 1.086 | 1.087 | 1.1147 | 1.114 | 1.115 |
| 5 | 1.0789 | 1.078 | 1.079 | 1.1094 | 1.109 | 1.110 |
| 6 | 1.0745 | 1.074 | 1.075 | 1.1021 | 1.102 | 1.103 |
| 7 | 1.0694 | 1.069 | 1.07 | 1.0995 | 1.099 | 1.100 |
| 8 | 1.063 | 1.063 | 1.063 | 1.0953 | 1.095 | 1.096 |
| 9 | 1.0538 | 1.053 | 1.054 | 1.0895 | 1.089 | 1.09 |

