# OpenReview forum: "Learning an Inventory Control Policy with General Inventory Arrival Dynamics"
_ICLR.cc/2024/Conference — Submitted to ICLR 2024_

### Official Review · Reviewer_8tKf · 2023-11-01

**Soundness:** 2 fair
**Presentation:** 1 poor
**Contribution:** 2 fair
**Rating:** 3
**Confidence:** 2

**Summary:**

- This paper proposes new quantity over time (QOT) inventory arrivals model instead of previously used vendor lead time (VLT) model.
- Other than the newly proposed inventory arrivals model, the paper seems to mostly follow the formulation of Madeka et al. (2022).
- The paper evaluates the proposed arrivals model with various experiments.

**Strengths:**

- Discusses an important but overlooked problem of inventory management
- Proposes various experiment settings to demonstrate the performance of proposed method

**Weaknesses:**

- The paper does not seem to be well presented. It is hard to follow, and majority of contents up to section 3 does seem to be overlapping with Madeka et al. (2022). Since the main contribution of the paper is about the QOT arrivals model, I believe that the RL part should not be  heavily described in the paper.
- The design of GenQOT model and the experiments should be explained in separate sections. Furthermore, I believe that the model design and learning methods should be detailed in the main text, not deferring it to appendix, since those are what first proposed in this paper.
- As GenQOT arrivals model is first proposed, I would expect other models, e.g. lead time model, to be compared with GenQOT in terms of their predictive performance. However, as far as I understood, the absolute errors of the proposed model is shown, and it is hard to judge whether the proposed model is useful or not.
- While the discounted reward is shown in table 2, I would expect GenQOT based agent to obviously outperform other agents when evaluated under Gen-QOT dynamics, as other agents are needed to adapt. Combined with the issue above, I believe that the performance of proposed model is not well proven overall.

**Questions:**

- Other than the RL experiment that seems to be obvious (as pointed out above), from what aspect do you think QOT or GenQOT is better than the other approaches?

---

### Official Review · Reviewer_3shQ · 2023-11-06

**Soundness:** 2 fair
**Presentation:** 2 fair
**Contribution:** 2 fair
**Rating:** 3
**Confidence:** 2

**Summary:**

This paper utilizes RL to solve inventory control problems. The paper formulates the inventory control problem as an exogenous decision process and designs GEN-QOT model to solve it. The empirical studies show the importance of an accurate model of the arrival dynamics.

**Strengths:**

The paper considers a problem motivated from real inventory control problems.  The formulation of the problem is sound.

**Weaknesses:**

The paper considers a concrete application and apply RL to solve it. I am not familiar with the application of inventory control, but the paper may have the following weakness.
- The paper seems to directly apply RL to solve the inventory control problem. It is not clear what are the challenges in solving the problem and how the method is improved comparing with existing methods.  If I do not miss anything, the novelty looks limited.

-  It is not clear what concrete RL algorithm is applied in this problem. Is the algorithm on-policy or off-policy? How the RL model is trained?

- The experiment results does not give enough insights on the effectiveness of the proposed model.  The authors may need to clarify the insights further.

**Questions:**

See the questions in the weakness session.

---

### Official Review · Reviewer_uujJ · 2023-11-12

**Soundness:** 2 fair
**Presentation:** 2 fair
**Contribution:** 2 fair
**Rating:** 3
**Confidence:** 2

**Summary:**

The paper applies RL to solve for an inventory system with lead time. Comparing to previous work (Madeka et al., 2022), the paper consider a different model where the orders arrive in multiple shipments instead of one whole shipment. The paper presents simulation results on their model evaluation as well as the policy performance.

**Strengths:**

The paper generalizes the arrival dynamics of the inventory system and provides simulation results to support their method.

**Weaknesses:**

The major contribution of the paper, compared to (Madeka et al., 2022), is to assume multiple shipments instead of one shipment. However it is unclear how significant such a contribution is:
1. Is the inventory system with multi-arrivals important to study? The authors should provide more related literature and real-world applications to the model that they consider.
2. How does the technique in solving the problem different from (Madeka et al., 2022) after introducing the additional structure in the model?

Since the paper is trying to learn an inventory system and obtain the optimal policy from a historical data set, it would be beneficial to discuss its relation to the offline reinforcement learning literature, which has established efficient algorithms with performance guarantees that is applicable to inventory systems:
1. Sidford, Aaron, et al. "Near-optimal time and sample complexities for solving Markov decision processes with a generative model." Advances in Neural Information Processing Systems 31 (2018).
2. Li, Gen, et al. "Breaking the sample size barrier in model-based reinforcement learning with a generative model." Advances in neural information processing systems 33 (2020): 12861-12872.

**Questions:**

In Section 3.2 the paper uses $\theta$ to parameterize the policy class, but it is unclear how $\pi$ is parameterized by $\theta$ and what type of policy is considered.

---

### Meta-Review · Area_Chair_iD3A · 2023-12-09

**Metareview:**

All reviewers question the novelty of the work as the innovation beyond Madeka et al., 2022 is limited. This clear weakness precludes the paper from being publishable by ICLR standards.

**Justification For Why Not Higher Score:**

N/A

**Justification For Why Not Lower Score:**

N/A

---

### Decision · Program_Chairs · 2024-01-16

Reject